# The Chloroplast *Trans*-Splicing RNA–Protein Supercomplex from the Green Alga *Chlamydomonas reinhardtii*

**DOI:** 10.3390/cells10020290

**Published:** 2021-02-01

**Authors:** Ulrich Kück, Olga Schmitt

**Affiliations:** Allgemeine und Molekulare Botanik, Faculty for Biology and Biotechnology, Ruhr-Universität Bochum, Universitätsstraße 150, 44780 Bochum, Germany; olga.reifschneider@rub.de

**Keywords:** group II intron, *trans*-splicing, ribonucleoprotein complex, chloroplast, *Chlamydomonas reinhardtii*

## Abstract

In eukaryotes, RNA *trans*-splicing is a significant RNA modification process for the end-to-end ligation of exons from separately transcribed primary transcripts to generate mature mRNA. So far, three different categories of RNA *trans*-splicing have been found in organisms within a diverse range. Here, we review *trans*-splicing of discontinuous group II introns, which occurs in chloroplasts and mitochondria of lower eukaryotes and plants. We discuss the origin of intronic sequences and the evolutionary relationship between chloroplast ribonucleoprotein complexes and the nuclear spliceosome. Finally, we focus on the ribonucleoprotein supercomplex involved in *trans*-splicing of chloroplast group II introns from the green alga *Chlamydomonas reinhardtii*. This complex has been well characterized genetically and biochemically, resulting in a detailed picture of the chloroplast ribonucleoprotein supercomplex. This information contributes substantially to our understanding of the function of RNA-processing machineries and might provide a blueprint for other splicing complexes involved in *trans*- as well as *cis*-splicing of organellar intron RNAs.

## 1. Introduction

One of the unexpected and outstanding discoveries in 20th century biology was the identification of discontinuous eukaryotic genes [1,2]. Walter Gilbert introduced the terms “exon” for expressed sequences and “intron” for intervening sequences [3]. In order to fuse exons and generate mature messenger RNA (mRNA), introns are posttranscriptionally removed by splicing. Several studies revealed that splicing in nuclei of eukaryotes is mediated by highly dynamic ribonucleoprotein machinery called the spliceosome [4]. The intervening sequences in the nuclear precursor mRNA of eukaryotes are therefore described as spliceosomal introns or nuclear mRNA introns. Further intron types have been identified and are classified by their structural characteristics and splicing mechanisms, i.e., archaeal introns, group I introns, and group II introns. Spliceosomal introns and group II introns share a common splicing mechanism (as outlined later) that is distinctly different from the splicing mechanism of group I introns, which are found in genomes of bacteria and organelles as well as in nuclei of lower eukaryotes. Finally, archaeal introns are comparably small and share a pre-tRNA intron excision mechanism with eukaryotes.

Here, we summarize our current knowledge about the relationship between group II introns and spliceosomal introns. This includes a brief description of introns-early and introns-late theories, a comparison of sequence homologies, and discussion of mechanistic similarities. We describe nuclear and organellar splicing factors further and discuss how *trans*-spliced and degenerated group II introns may have evolved into spliceosomal introns.

In the second part of our review, we briefly discuss *trans*-spliced introns in different organismal lineages and finally summarize the extensive experimental data supporting the existence of a *trans*-splicing supercomplex in *C. reinhardtii* chloroplasts.

## 2. Introns-Early or Introns-Late Debate

Very soon after the discovery of spliceosomal introns, several theories arose about their origin and evolution. The introns-early theory published around the same time is closely coupled to the exon theory of genes [5,6,7]. The main hypothesis is that the split structure of genes is ancient and occurs in all domains of life, and that exons are ancestral genetic elements encoding proteins with small domains. Thus, multidomain proteins evolved by recombination of neighboring exons that were connected by noncoding sequences [8,9]. Subsequently, these noncoding sequences became introns. However, further investigations showed inconsistent correlation between exons and protein domain units [10].

An alternative, the introns-late theory, was proposed by different authors [11,12]. According to this hypothesis, spliceosomal introns are unique to eukaryotes, introduced into the host genome by an endosymbiont (Figure 1). The endosymbiosis of α-proteobacteria entailed massive genomic transfer accompanied by invasion of the host genome with group II introns. This type of intron has the ability to self-splice and invade DNA, and it is found in abundance in eubacterial genomes [13,14]. Strong selective pressure on the archaeal host may have initiated the fragmentation of group II intron RNA into five small nuclear RNAs (snRNAs) and the mRNA intron, which now function in splicing of spliceosomal introns [15]. Consistent with this idea of intron evolution is the distribution of group II introns [13]. They are frequently found in bacterial genomes and were most likely introduced by horizontal transfer to eukaryotic nuclei [16].

So far, group II introns have not been found in nuclear genomes of eukaryotes, which indicates that they were efficiently transformed into spliceosomal introns during evolution. The introns-late theory was further expanded by proposing that the invasion of group II introns was one of the major driving forces for nucleus–cytoplasm compartmentalization [17]. Accordingly, the “turbulent phase” of gene transfer, including the spread of group II introns throughout the host genome, was followed by the development of low-efficiency splicing reactions. These circumstances led to the accumulation of unspliced precursor RNAs and translation of non-sense proteins. Thus, the formation of a nuclear envelope separated RNA splicing and protein translation within the cell.

Overall, the evolutionary development of spliceosomal introns delineated in the introns-late theory seems to be most likely. However, more parallels exist between group II introns and their putative counterparts in the nuclei of eukaryotes, as we discuss in the following sections.

## 3. Group II Introns and Spliceosomal Introns Share Identical Splicing Mechanisms

One of the most striking parallels of spliceosomal and group II introns is their identical splicing mechanism [11,18]. Current textbooks describe in detail the two transesterification reactions that occur during RNA splicing of group II introns. One important intermediate arising during this process is a lariat-formed RNA molecule that resembles the intron stem-loop lariat catalyzed by the spliceosome during nuclear pre-mRNA splicing. Thus, the formation of an intron lariat is a conserved feature of both the spliceosomal and group II intron splicing reaction.

In vitro, the splicing mechanism seems to be a simple chemical reaction, but in vivo, the process requires an intricate network of RNA–RNA interactions which are stabilized by protein factors [14,19,20,21]. The initial step during splicing is the recognition of exon–intron boundaries and the branch point nucleotide. Next, the 5′ss, 3′ss, and branch point are juxtaposed to create the catalytic core, where they are surrounded by further catalytic RNA structures and become reactive. These steps are performed by conserved RNA sequences and structures.

## 4. Sequence Homologies and Mechanistic Similarities

All group II introns display a characteristic and highly conserved secondary structure, (Figure 2). A central core containing the splice sites is surrounded by six helical domains (DI-DVI). Sequence similarities are rare and only occur in catalytic structures [22]. RNA–RNA interactions between intron sequences assemble a conserved tertiary structure that forms the catalytic core [23,24]. Deletion analysis showed that minimal catalytic activity requires at least DI and DV [25]. DII and DIII enhance activity but are not essential for accurate splicing [26,27]. Finally, DIV harbors an optional open reading frame (ORF) encoding a maturase [28]. These proteins comprise three distinct domains: an RNA-binding and splicing domain (X), a reverse transcriptase (RT) domain, and a DNA-binding/endonuclease (D/En) domain. The latter mediates the mobility of group II introns.

In contrast to group II introns, splicing of nuclear mRNA introns requires the intron sequence itself and *trans*-acting RNAs. The pre-mRNA intron harbors the 3′/5′ss and the branch point nucleotide. However, besides the intron sequence, splicing of nuclear mRNA introns requires five *trans*-acting RNAs, the snRNAs. The snRNAs assemble with protein factors and form ribonucleoprotein (RNP) complexes [29]. For each splicing reaction, these RNPs associate de novo on one intron together with a further set of more than 100 proteins and the intron RNA [19]. This highly dynamic RNP splicing machinery is known as the spliceosome.

According to the introns-late theory, snRNAs originated from group II intron sequences. Indeed, similar to the helical domains of group II introns, they mediate exon/intron recognition and catalytic core formation [30].

The detection of introns is also crucial for splice site recognition, mainly because they are located directly at the boundaries of exon sequences. Consistent with the introns-late theory, group II introns and spliceosomal introns share sequence similarities at their intron ends. Group II introns start with 5′-GUGYG and end with AY-3′ (Y = C, U), while nuclear U2-type mRNA intron boundaries comprise 5′-GU-AG-3′ [14]. The structural similarities between group II introns and spliceosomal introns and their similar chemistry of splicing were excellently discussed recently [31,32].

Besides recognizing regions defining exon and intron boundaries, efficient splicing of group II and nuclear mRNA introns requires a bulged nucleotide, which is involved in the first nucleophilic attack. With group II introns, this nucleotide is generally an adenosine located in domain DVI (Figure 2). In spliceosomal introns, the branch point emerges close to the 3′ss after binding of U2 snRNP to the mRNA intron sequence [33,34]. This recognition is maintained by a consensus sequence and binding results in a single bulged nucleotide. The common fixation of a bulged nucleotide conformation is a further indication for the evolutionary relationship of both intron types [35].

The catalytic core is the most striking parallel between group II and spliceosomal introns [23,36,37]. It requires the 3′/5′ss and branch point nucleotide to be juxtaposed in a reactive manner and is characterized by further catalytic RNA structures and sequences [36,38]. In group II introns, the catalytic core is defined by the interaction of two distinct structures, the junction between DII and DIII (J2/3) and DV [14,39] (Figure 2). DV is the most phylogenetically conserved structure in group II introns and, therefore, is often described as the heart of a group II intron [38,40]. It comprises two short helices separated by a bulged region and contains the catalytic triad (5′-RGC, R = A/G) crucial for binding of J2/3. This interaction results in a triple helix structure [41]. The DV bulge (Figure 2) is mainly responsible for metal ion binding, which is required for efficient splicing [35,41].

The equivalents in the spliceosome are found in the U6/U6atac snRNAs [37,42,43]. The evolutionary conserved elements are the ACAGAGA box corresponding to J2/3 and an AGC triad in a bulged internal stem loop that corresponds to domain DV. The importance of these structures was supported by mutation analyses of the corresponding sequences, resulting in complete or partial loss of splicing efficiency [44,45,46]. These remarkable similarities in RNA structure and functionality provide further hints that snRNAs originated from group II introns.

## 5. *Trans*-Spliced and Degenerated Group II Introns: A Step Toward Spliceosomal Introns?

The origin of *trans*-acting snRNAs from fragmentation of ancestral group II introns is a reasonable theory to explain the evolution of spliceosomal introns. However, neither intermediate stages of this fragmentation process nor sequences encoding group II introns have been identified so far in more than 1000 sequenced nuclear genomes [14,47,48]. Recently, group II introns artificially inserted into nuclear genes of *Saccharomyces cerevisiae* were shown to be expressed and spliced [49,50]. However, the group II introns were not spliced until they were transported into the cytoplasm, in contrast to spliceosomal introns that are spliced exclusively in the nucleus. Additionally, translation of group II intron-spliced mRNA was inhibited due to translational repression.

The search for intermediate states in the transition of group II introns to nuclear mRNA introns therefore focused on cell compartments (chloroplasts, mitochondria) and bacteria that still contain group II intron-encoding sequences. Indeed, discontinuous group II introns were identified in organellar genomes and were used to illustrate what could have occurred during evolution [15,47,51,52]. As a consequence of genome rearrangements, exons of genes encoding such fragmented group II intron structures are dispersed across the eukaryotic genome [53]. The independently transcribed precursor RNAs associate through base pairing and tertiary interactions to generate the catalytic conserved group II intron structure. Therefore, mature mRNA is generated by *trans*-splicing, in contrast to the process of *cis*-splicing that involves a single continuous RNA precursor.

The first organellar *trans*-spliced group II introns were identified in the late 1980s [54,55,56]. The majority of *trans*-split group II introns are dipartite and disrupted in domain DIV [52,57]. This domain is insufficient for proper splicing, but it harbors an optional ORF encoding a maturase (Figure 3A). The fragmentation is consistent with the observation that nearly all organellar group II introns have lost the maturase-encoding gene [20,58]. If *trans*-spliced group II introns still harbor an ORF, a fragmentation site is located either upstream or downstream of the ORF [59]. Hence, fragmentation of the group II intron structure in DIV occurs quite often, while other domains are less affected (Figure 3C).

Two dipartite group II introns were shown to be disrupted in DIII and are located in the rps12 gene of Marchantia and Nicotiana [54,61]. Additionally, the rbcL-intron 2 (rbcL-i2) of the green algae *Floydiella terrestris* and *Stigeoclonium helveticum* is disrupted in DII [62,63]. RNA elements of the helical DII and DIII structures are also not sufficient for splicing [64]. Furthermore, some dipartite group II introns (rbcL-i1, psaC-i1, and petD-i1) in plastids of a few green algae show fragmentation in DI [62,63,65].

Additional fragmentation of group II introns leading to a tripartite structure is rare, with two representative examples: the chloroplast intron psaA-i1 in the green alga *Chlamydomonas reinhardtii* and the mitochondrial intron nad5-i4 in the flowering plant *Oenothera berteriana* [56,66]. Both introns exhibit one disruption in DIV and one in DI, which occurs close to the ε-binding site (Figure 2). Thus, the EBS1/δ-containing structure of domain DI, DII, and DIII must be reconstituted by a third RNA fragment. In both organisms, loci encoding such *trans*-acting RNA structures have been identified and are located in distant genomic regions. These are the tscA (*trans*-splicing of chloroplast psaA mRNA) locus in *C. reinhardtii* and the tix (*trans*-splicing intron fragment) locus in *O. berteriana* [66,67]. Surprisingly, structural predictions indicate that the tscA RNA provides only a degenerated DI lacking EBS1 and δ-interaction sites. The absence of these structures important for the exon recognition process leads to the question of whether a fourth RNA is involved in psaA-i1 splicing [67].

The ability of natural continuous group II introns to *trans*-splice was previously investigated using a transposon-based genetic screen [51,68,69], where dipartite and tripartite versions of the Ll.LtrB group II intron from *Lactococcus lactis* were generated to analyze their capacity to splice. The Ll.LtrB intron proved to be remarkably tolerant to fragmentation and able to splice when such fragmentation corresponded to naturally occurring sites in DI, DIII, and DVI. In contrast, fragmentation was not tolerated within functionally important elements, such as EBS1/EBS2, DVI, and structures involved in central core formation, including the splice sites.

In a recent approach, an alternatively spliced group II intron was even identified in the bacterial pathogen *Clostridium tetani* [70]. The C.te.I1 intron is located in the surface layer gene (SLP) and is able to undergo four alternative splicing reactions in vivo, thus producing different isoforms of SLP necessary for virulence and resistance to the host immune response. Alternative splicing is common in spliceosomal introns. Approximately 95% of human genes are thought to be alternatively spliced [71]. This process leads to a dramatic increase in information encoded by the genome [72,73]. The fact that group II introns are also used in this advantageous manner emphasizes their functional similarities with spliceosomal introns.

Another process indicating a transition toward nuclear mRNA introns includes deletions within the intron sequence. All mitochondrial and plastid group II introns of higher plants have lost the ability to self-splice, in contrast to bacterial group II introns [74]. This is primarily due to a degenerated or completely lost maturase gene in DIV [75,76]. However, several other alterations have been observed. For example, some mitochondrial group II introns in land plants display an abnormal DVI structure and lack a bulged adenosine, as was shown for nad1-i1, nad1-i2, nad4-i2, rpl2, and rps3 [47,77]. The homologous intron of nad4-i2 in the green alga *Chara vulgaris* exhibits a common DVI structure [78]. This emphasizes that alterations in higher plants occurred by rearrangements of a classical group II intron structure.

The occurrence of split group II introns in organellar eukaryotic genomes demonstrates that degeneration and fragmentation is possible and might represent an intermediate stage in the evolution of spliceosomal introns. Furthermore, the identification of *trans*-acting RNAs during organellar group II intron splicing illustrates how spliceosomal snRNAs may have originated.

## 6. Splicing Factors: Ribozymes Need Facilitators

Group II introns are ribozymes with their own enzymatic activity. Thus, intron RNAs are able to catalyze their own splicing reaction. Moreover, self-splicing was demonstrated under nonphysiological conditions and in the absence of protein cofactors [79,80,81]. Nuclear mRNA introns are spliced by complex RNP machinery, requiring both protein and RNA components, although considerable speculation remains as to whether the excision of spliceosomal introns depends solely on the catalytic activity of RNAs (mRNA intron and snRNAs). Recent studies revealed that a protein-free RNA complex comprising U6 and U2 snRNAs was able to perform both steps of the splicing reaction [82,83,84]. This provides evidence that nuclear mRNA splicing may also be based on catalytic RNAs. However, in vivo splicing of both intron types requires *trans*-acting protein factors [19,20,47]. Since they are not directly involved in the enzymatic activity of intron excision, these proteins must have other important functions.

## 7. Evolution and Function of Nuclear Splicing Factors

According to the introns-late theory, mobile group II introns from a bacterial endosymbiont invaded the host genome and evolved into nuclear introns and snRNAs. The mobility and the splicing reaction of these ancient introns depend on the intron encoded maturase gene. Thus, the invasion also introduced the first splicing factor into the host cell. Although not detected for a long time, extensive analyses of sequence and structure similarities eventually led to the discovery of splicing factor Prp8 (pre-mRNA processing 8), which displays significant homology to group II intron maturases [31,60] (Figure 3).

The discovery of Prp8 was the first proof for the origin of a spliceosomal protein from ancestral invaders. Moreover, Prp8 is the most conserved eukaryotic splicing factor, with more than 60% homology between fungi and mammals or plants [60]. The 280 kDa protein is closely associated with the catalytic core of the spliceosome through direct interaction with the splice sites, the branch point, U2, U5, and U6 snRNAs [30,85]. Its main function is the formation of catalytic structures, while the splicing activity itself depends on RNAs [84,86]. The conversion of a bacterial maturase into a core splicing protein presumably required the degeneration of further maturase genes and expansion of the targeted introns, either before invasion or after integration into the host genome [28]. Nevertheless, this evolutionary link does not explain how RNP machinery as complex as the spliceosome could evolve.

The most likely scenario is the recruitment of host or bacterial genes after endosymbiosis and their adaptation to the splicing process. One example of such a putative adaptation event is provided by the Sm/Lsm (Sm-like) protein family [87]. Sm proteins are abundant in the nuclear splicing machinery, where they assemble as hetero heptamers to form a ring structure around snRNAs, and thus stabilize the RNA [88,89]. Lsm proteins play a more general role in RNA processing, including chaperoning, degradation, translation, and also splicing [90]. Counterparts of these proteins also exist in archaea (Sm proteins) and bacteria (Hfq protein), where they function in RNA processing [91,92]. The functional and structural similarities provide evidence that the Sm/Lsm proteins might have been recruited as splicing factors at early stages of spliceosome development [87].

Another hypothesis proposes that during spliceosome evolution, functional elements of group II introns were successively replaced by de novo protein components [30]. An example for such a process is the protein-assisted branch point formation in the spliceosome. In contrast to group II introns, where the bulged adenosine is anchored in the DVI stem loop, spliceosomal introns must interact with U2 snRNP to generate a bulge [34,93]. This interaction is initiated and stabilized by several RNA-binding proteins that are abundant in the spliceosome.

The main function of RNA-binding proteins is to stabilize RNA structures. The majority of spliceosomal RNA-binding proteins belong to the SR protein family, which possess one or two copies of an N-terminal RRM (RNA recognition motif) domain and a serine-/arginine-rich (RS) C-terminal domain. Both structures can bind RNA [30,94]. The RRM domain is not restricted to SR proteins; it is also present in other splicing factors. For example, Prp24 displays four RRM domains crucial for binding to U6 snRNA [95]. Furthermore, the splicing factors U1-A and U1-70k harbor an RRM domain, and Prp8 described above contains a degenerated copy [86,96]. Further RNA-binding domains have been identified, such as the RNA-binding domain of U2AF or the S1 binding motif of Prp22 [30,97].

Apart from stabilization, many splicing factors function as helicases that remodel RNA–RNA and RNA–protein interactions. In eukaryotes, many helicases involved in splicing are well conserved [98]. They are able to unwind RNA structures and thus initiate conformational rearrangements [99].

By stabilizing RNA structures and remodeling interactions, nuclear splicing factors regulate and coordinate the splicing reaction and help the RNA display its catalytic conformation. While Prp8 probably derived from an ancestral maturase gene, the majority of nuclear splicing factors evolved de novo or were recruited from the host or bacterial endosymbiont and adapted to assist in splicing.

## 8. Splicing Factors in Plastids and Mitochondria

Organellar group II introns differ significantly from their bacterial ancestors. Degeneration and fragmentation occurred frequently in organellar group II introns, while bacterial group II introns remained nearly as intact as their ancient progenitors. The loss of self-splicing and a group II intron-encoded maturases suggest that they became more dependent on *trans*-acting factors. Apart from maturases, further protein homologues of the nuclear spliceosome machinery have indeed not been observed to date. Nevertheless, 20 years of research on organellar splicing factors has revealed that convergent evolutionary processes occurred. They initiated the evolution of *trans*-acting splicing factors related to their counterparts in the nucleus in order to adapt to fragmented and degenerated intron RNA.

Along with the green alga *Chlamydomonas reinhardtii*, the organelles of higher plants are useful sources to study group II intron splicing factors. *Arabidopsis* and *Zea mays*, for example, harbor approximately 20 group II introns in both mitochondria and chloroplasts. None of them has been shown to self-splice in vitro, and only a single intron in each organelle has retained the maturase-encoding ORF, namely the plastid matK positioned in the trnK group II intron, and the mitochondrial matR gene located in nad1-i4 [28,100,101,102] (Figure 3). The absence of further maturases implies that MatK and MatR might target multiple introns. Indeed, both maturases are able to bind several different group II introns in the corresponding organelles [28,103] Furthermore, a homologue of matK still exists and is expressed in organisms that have lost the trnK gene, and with it, the corresponding group II intron [104,105].

In addition, four further maturase genes have been identified in nuclear genomes of land plants (nMat1-4; [106,107]). Whereas nMat4 might be localized in both plastid and mitochondrial organelles, nMat1-3 are predicted to be solely imported into mitochondria [28,108,109]. Recent studies also indicate a splicing function of a subset of group II introns for all nMats [47]. Together, this provides evidence that intron-specific maturases were converted into general splicing factors in organelles.

Splicing of group II introns in organelles depends on further nucleus-encoded *trans*-acting factors, which lack homologies to maturases [20,47,58]. While some assist in splicing of a single group II intron, the majority of these factors target multiple introns. For example, CRS2 (chloroplast RNA splicing 2) and RNC1 (Ribonuclease 1) of *Zea mays* chloroplasts were shown to target nine different introns, and Raa1 assists in splicing of both *C. reinhardtii* group II introns [110,111,112]. Their function in splicing is thought to have evolved in the same way as in the nucleus: either by de novo evolution or by recruitment from existing RNA metabolism-related proteins that resemble bacterial proteins [113]. CRS2, for example, shows homologies to bacterial peptidyl tRNA hydrolase, and Raa2 from *C. reinhardtii* shares similarities with TruB-like pseudouridine synthases, also found in bacteria [110,114].

Group II intron splicing depends on RNA-binding proteins with diverse RNA-binding motifs [115]. A frequently occurring domain in land plant splicing factors is the CRM (chloroplast RNA splicing and ribosome maturation) domain [116,117,118]. This CRM domain is structurally related to the above-mentioned RRM domain involved in nuclear pre-mRNA splicing, providing evidence for a common mechanism of RNA recognition [119]. A further, greater number of organellar splicing factors belong to the helical repeat superfamily, such as PPRs (pentatricopeptide repeat), and mTERFs (mitochondrial transcription termination factor), which commonly exhibit tandem repeats of a conserved motif [47,120]. PPR proteins in land plants are believed to bind RNA in a sequence-specific manner, and thus mostly target single group II introns [20,121]. In *C. reinhardtii*, the plastid splicing apparatus lacks PPR proteins, but contains OPR (octatricopeptide repeat) proteins, an additional subgroup of the helical repeat superfamily [122,123].

Organellar splicing factors likely stabilize the RNA and help it fold into catalytically active structures by binding its target RNAs. This assumption is supported by analysis of the splicing factor CRS1 from *Zea mays*, which promotes folding of group II intron domains [124]. Furthermore, the splicing factor PMH2 (putative mitochondrial helicase 2) from *Arabidopsis* belongs to the DEAD-box helicase family and is involved in multiple splicing reactions. Corresponding mutants display reduced splicing efficiency with several group II introns in mitochondria [125].

A common feature of the spliceosome and organellar splicing factors is the formation of high molecular weight complexes [120,126,127,128]. However, all U2-type introns in the nucleus are spliced with the help of RNPs, which consistently display a similar composition. In contrast, organellar complexes are diverse and contain both general and intron-specific protein factors, and thus their composition differs depending on the target intron. Nonetheless, evidence exists for spliceosome-like machineries in organelles that specifically act on single introns.

In contrast to group II introns of organelles, splicing of bacterial group II introns only depends on the intron-encoded maturase. These maturases form RNPs with corresponding introns (Figure 3). The *Lactococcus lactis* group II intron Ll.LtrB forms an RNP with a dimer of its maturase LtrA [129,130]. By binding the intron, the maturase stabilizes the active structure, which can rapidly revert when LtrA is digested by a protease [131,132]. In vitro analyses with recombinant LtrA and the intron RNA revealed that this minimal RNP formation is sufficient and necessary for the splicing reaction [133,134].

## 9. Occurrence of *trans*-Splicing in Organellar Genomes

In the 1980s, the completely sequenced chloroplast genomes from liverwort and tobacco provided the first evidence for *trans*-spliced introns (reviewed in [52]). Further evidence was provided when mature mRNAs derived from independently transcribed precursor RNAs were detected by heteroduplex analysis and cDNA sequencing of the rps12 mRNA from tobacco, and cDNA sequencing of the psaA mRNA from the green alga *C. reinhardtii*. Subsequent DNA and RNA sequencing data from a wide range of organisms showed that organelle *trans*-splicing occurs in almost all clades of life (extensively reviewed in [52]). Since then, extensive analyses of mitochondrial and plastid genomes have identified representatives in diverse organisms [52,58].

*Trans*-spliced group II introns are frequently found in angiosperms, with up to six known examples in mitochondrial NAD genes encoding subunits of NADH dehydrogenase [47,135]. Recently, the mitogenomes from 15 diverse gymnosperms were determined [136]. Interestingly, most exons from protein encoding genes are distantly dispersed on the mtDNA. RNA-Seq analysis confirmed that 50% to 70% of all mature transcripts are generated by RNA *trans*-splicing.

During evolution, mitogenomic rearrangement of mitochondrial DNA seems to be responsible for multiple *trans*-splicing events in the vascular plant lineage [137]. *Trans*-splicing, as described above, is usually catalyzed by group I or group II introns. It is distinctively different from *trans*-splicing observed for diplonemide mitogenomes [138]. Diplonemids are heterotrophic marine flagellates and their mtDNA comprises a number of covalently closed circular chromosomes. The circular molecules all carry short unique cassette regions, which are separately transcribed. Mature transcripts are generated by *trans*-splicing of precursor RNAs, yielding mature RNA (mRNA or rRNA). However, the mechanism behind this process remains to be solved [139].

## 10. Two Group II Introns of the Chloroplast psaA Gene Are *trans*-Spliced in *C. reinhardtii*

To date, the best-analyzed *trans*-splicing process is the posttranscriptional processing of psaA mRNA in the unicellular green alga *C. reinhardtii*. As outlined below, the investigation of a *trans*-splicing complex led to the discovery of splicing machinery that requires RNP formation and depends on dynamic interactions. This is reminiscent of RNP-dependent association of catalytic structures during splicing of spliceosomal introns. Thus, the comparison of homologies between group II and spliceosomal introns hint at several evolutionary routes through which self-splicing ribozymes were converted into a protein-dependent splicing apparatus.

*C. reinhardtii* can be considered the model organism for analyzing plastid gene expression and communication between the nucleus and the chloroplast. *C. reinhardtii* carries two chloroplast group II introns that are both part of the psaA gene. The two introns are fragmented and dispersed within the chloroplast genome [55,67]. The three exon precursors of psaA are transcribed independently to generate unspliced RNA molecules, and two *trans*-splicing reactions are required to generate the mature psaA mRNA. For intron 1, three independently transcribed RNA molecules assemble into a functional group II intron structure by base pairings and tertiary interactions (Figure 2). This tripartite group IIB intron is interrupted in domains DI and DIV; thereby, exon 1 is flanked by a portion of domain DI and exon 2 by the entire domains DIV+V as well as a portion of DVI. The rest of domain DI and DIV as well as the entire DII+III domains are delivered from a third RNA molecule, the plastid-encoded tscA RNA, which is 450 nt in length. Homologous sequences to tscA were also detected in the plastomes of other *Chlamydomonas* species [52]. A recent deep RNA-Seq study showed that 71.7–96.7%, of the RNA transcripts were in the spliced form, and almost no differences were observed between culture conditions, such as light versus dark or plus versus minus Fe [140].

## 11. Nuclear-Encoded Proteins Promote the Chloroplast *trans*-Splicing Process

mRNA maturation of the tripartite chloroplast psaA gene depends on various nucleus-encoded factors that participate in *trans*-splicing of the two group II introns. A thorough genetic analysis of photosynthesis mutants identified many strains that also have a defect in *trans*-splicing of pre-mRNA [111,141]. The corresponding mutants were abbreviated as RAA (RNA maturation of psaA RNA) or RAT (RNA maturation of psaA tscA RNA). In the meantime, further major psaA *trans*-splicing factors have been structurally and functionally characterized and are mentioned throughout this review (listed in Table 1).

## 12. The *Chlamydomonas* Chloroplast Spliceosome

Recently, a multiprotein complex was identified that is involved in processing the psaA precursor mRNA. Using coupled tandem affinity purification (TAP) and mass spectrometry analyses with multiple *trans*-splicing factors as bait proteins, a multisubunit ribonucleoprotein (RNP) complex was characterized comprising the previously characterized *trans*-splicing factors Raa1, Raa3, Raa4, and Rat2 plus novel components [143]. Raa1 and Rat2 share the structural octatricopeptide repeat (OPR) motifs that presumably function as RNA interaction modules. Two of the novel RNP complex components also exhibit a predicted OPR motif and were therefore considered potential *trans*-splicing factors.

To identify defective genes involved in particular *trans*-splicing mutants, functional complementation assays were conducted using bacterial artificial chromosome (BAC) vectors. This assay, for example, revealed that the *trans*-splicing defect of mutant F19 was restored by a new factor, called RAA8 [123]. Molecular characterization of complemented strains verified that Raa8 participates in splicing of the first psaA group II intron (psaA-i1). Three of six OPR motifs are located in the C-terminal end of Raa8, which was shown to be essential for restoring psaA mRNA *trans*-splicing. This result supports the important role played by OPR proteins in chloroplast RNA metabolism and also illustrates that TAP and mass spectrometry combined with functional complementation studies represents a vigorous tool for identifying *trans*-splicing factors. This method was first developed for yeast and has since been used in a variety of organisms to elucidate various interactomes [146,147,148,149,150,151].

Follow-up investigations implicated the involvement of RNP complexes in psaA pre-mRNA splicing, and thus the presence of a putative plastid spliceosome resembling the nuclear counterpart [126,127]. The plastid spliceosome concept gained ground when technically advanced methods further contributed to our understanding of the composition of a spliceosomal complex. These analyses indicated that at least two RNPs promote splicing of the psaA pre-mRNA.

TAP analyses revealed the core components of two subcomplexes required for psaA pre-mRNA splicing. Core proteins of subcomplex I were identified by TAP–MS with cultures grown under dark-adaptation and anaerobiosis. Comparison with previous results obtained with standard growth cultures revealed a protein cohort present under all tested conditions [123]. This protein cohort comprises ten subunits, including the factor Raa1 (splice defect in psaA-i1, and psaA-i2), and four splicing factors involved in splicing psaA-i1: Raa3, Raa4, Raa8, and Rat2 [111,123,126,142,145]. In a complementary approach, subcomplex I was previously purified with the second bait protein Rat2 using standard growth conditions, and data sets of Raa4::TAP and Rat2::TAP were compared [123]. Remarkably, this complementary approach revealed nearly the same set of subcomplex I core components as obtained with cultures grown under alternative conditions [143]. Nine of the ten identified protein subunits of subcomplex I, including all known splicing factors (Table 1), were found with both approaches. Thus, both strategies seem to be powerful tools to discover true complex components.

Furthermore, it was shown that subcomplex I is an RNP since it can be copurified with exon 1 precursor and the tscA RNA, both necessary for psaA-i1 formation [123]. In addition, the size of subcomplex I was analyzed by size exclusion chromatography (SEC). A Raa3-containing complex of 1.7 MDa was observed, whereas the detection of Raa4 and Rat2 revealed a putative complex with the size of 1.9 MDa [123,126]. The combined molecular mass of the ten subcomplex I core subunits identified in this study is 1.5 MDa. Adding together the psaA exon 1 precursor (120 kDa) and tscA RNA (130 kDa) gives a molecular mass of 1.75 MDa. This is consistent with SEC results, as mentioned above. Further evidence that subcomplex I contains RNA was provided by TAP analyses with Raa4 and Rat2 in a tscA deletion strain [123]. Deletion of tscA resulted in an altered subcomplex I that lacked several core components, such as Raa3. These results not only confirm that subcomplex I is an RNP, but also show that RNA is crucial for subcomplex I formation.

The identification of an RNP involved in the first psaA splicing reaction implied that a further subcomplex II probably mediates splicing of the second psaA intron. This hypothesis was supported by the identification of Raa1, a general splicing factor that functions in both psaA splicing reactions [111]. Furthermore, characterization of mutant Δraa2 revealed splicing factor Raa2, with a defect in psaA-i2 splicing [114]. This was the first report of a protein that is essential for splicing of psaA-i2. Subcellular fractionation experiments showed that Raa2 and Raa1 colocalize in a 500 kDa complex, but its precise composition remained unclear [127]. TAP–MS with Raa2 as bait revealed putative subcomplex II components, confirming that both Raa1 and Raa2 are part of the same complex. Furthermore, a third, recently characterized splicing factor Raa7 was copurified with Raa2. Using TAP–MS with Raa7 as bait revealed that the core of subcomplex II comprises at least seven subunits, including Raa1, Raa2, and Raa7 [144]. As confirmed by SEC, these core components have a combined mass of 500 kDa, and form subcomplex IIA. Association of subcomplex IIA with RNA corresponding to the second psaA group II intron (~1 MDa) results in the formation of subcomplex IIB (1.5–2 MDa). Indeed, enrichment of the psaA exon 2 and psaA exon 3 precursors was demonstrated previously by qRT-PCR [143]. However, SEC results indicate that subcomplex IIA (500 kDa) accumulates independently of subcomplex IIB (1.5–2 MDa). In contrast to subcomplex I, which associates in an RNA-dependent manner, subcomplex IIA (500 kDa) is presumably formed solely by protein factors before binding to psaA-i2 RNA. This idea is reinforced by previous sucrose gradient centrifugation experiments, which revealed that subcomplex IIA (500 kDa) is not sensitive to RNase treatment [127]. On the contrary, the complex formation is abolished in Δraa1 and Δraa7 mutants, indicating that Raa1 and Raa7 are essential components of subcomplex II (500 kDa) and prerequisites for complex formation [127].

The formation of RNPs obviously requires the presence of RNA-binding proteins. RNA-binding properties have been observed for several splicing factors in *C. reinhardtii* and land plants [142,152,153]. The RNA binding characteristics of a subcomplex II subunit were demonstrated by electrophoretic mobility shift assay (EMSA) analyses with a recombinant Raa2 protein, which revealed a direct interaction between Raa2 and DI-III of psaA-i2 [154]. DI is involved in recognition of exon/intron boundaries. Therefore, interaction with Raa2 might stabilize conserved domain structures of psaA-i2 and induce the RNA-RNA interactions required for exon recognition. However, to elucidate the function of this interaction, binding sites of Raa2 and other splicing factors have yet to be determined. The appropriate techniques for such studies are RNA footprint analysis, a powerful tool to characterize specific RNA sequences bound by a protein [155], or the CLIP (cross-linking and immunoprecipitation) approach, which has proven to be an efficient method to analyze RNA-binding sites of proteins [156].

Subcomplex I and II only share splicing factor Raa1. The functionality of Raa1 was analyzed using complementation analyses [111]. The C-terminal region is essential for splicing of psaA-i1. In contrast, the middle part of Raa1 is involved in splicing of psaA-i2, and the N-terminal sequence is not essential for splicing. Thus, Raa1 functions in both splicing processes, but seems to have specialized domains for each splicing reaction. These domains may harbor RNA-binding sites, such as OPR motifs, that specifically recognize the first or second psaA group II intron. Furthermore, these motifs can function in protein–protein interactions with the C-terminus of Raa1 and the middle part of Raa2 or Raa7 (Figure 4). However, it remains unclear whether a single Raa1 molecule functions simultaneously in both reactions or separately in each *trans*-splicing reaction.

Both subcomplexes of the psaA splicing machinery are copurified with the target intron RNA and thus display RNPs. Similarly, RNPs were also described for group II intron splicing of land plants [112,120,152,157,158]. Highly complex RNPs of land plant group II introns were described for ndhB, petB, petD, and trnG, with at least five splicing factors known for each intron [20,152].

## 13. Evidence for the Formation of a Supercomplex During psaA Pre-mRNA Splicing

The class A mutant Δraa6 shows a defect in the second psaA pre-mRNA splicing reaction, and characterization of this mutant revealed the splicing factor Raa6 [142]. Despite the function of Raa6 in the second reaction, this splicing factor was not copurified with Raa2 or Raa7 in TAP experiments. This observation implies that Raa6 is not a component of the RNA-independent subcomplex IIA (500 kDa). Remarkably, further mutants impaired in psaA-i2 splicing exist that show no alteration in the formation of subcomplex IIA (500 kDa). This was shown by the fractionation of crude protein extracts derived from selected class A mutants in a sucrose gradient and detection of the 500 kDa complex with a Raa2-specific antibody [127]. While Δraa1 and Δraa7 mutants showed complete abolition of this complex, photosynthetic mutants Fl38, HN12, and HN54 exhibit signals in 500 kDa fractions, similar to wild-type extracts. Although the loci affected in these three mutants are still undetermined, these results indicate that some splicing factors defective in psaA-i2 splicing are not involved in subcomplex IIA (500 kDa) formation but seem to have alternative functions.

To elucidate the role of Raa6 and related splicing factors (defect in psaA-i2 splicing), TAP–MS analyses with Raa6 as bait were performed [143], in addition to one-step purification using Raa6 as bait. The TAP approach revealed that Raa6 is co-purified with several proteins that were previously identified with Raa2, but not with Raa7 as bait (Figure 4). This is consistent with SEC experiments that revealed a colocalization of Raa2 and Raa6 in high molecular weight fractions of 1.5–2 MDa. This implies that Raa6 probably interacts with subcomplex IIB (1.5–2 MDa) via Raa2 or other proteins copurified with Raa2 and Raa6. Thus, Raa6 probably associates after or during interaction of subcomplex IIA (500 kDa) with the psaA-i2 intron RNAs leading to subcomplex IIB (1.5–2 MDa) and is not a part of the RNA-independent subcomplex II (500 kDa). In addition, both the tandem approach and one-step purification revealed that core components of subcomplex I and II were copurified with Raa6 (Figure 4). This includes Raa8, Raa4, the CLP protease, the U1-specific protein C of subcomplex I, Raa2, the plastid ribosomal protein 20, and the threonine dehydrogenase of subcomplex II [143]. These data suggest that both subcomplexes are connected and probably form a supercomplex. Since core components of subcomplex I and II were identified in low amounts and not in all replicates, such an interaction may be transient. Previous TAP experiments showed that Raa6 was copurified with an altered subcomplex I after deletion of the tscA RNA [123]. This deletion curtails psaA-i1, inhibiting the first splicing reaction. Raa6 was observed to accumulate with Raa8, Raa4, and Rat2, while Raa3 and further core components of subcomplex I were absent.

Thus, the one-step purification with Raa6 as bait shows similar results to Raa4 purifications in a ΔtscA background. Furthermore, yeast two-hybrid analyses confirmed direct protein–protein interactions between Raa6 and splicing factors of both subcomplexes [143]. These results provide further evidence for a putative interaction between both splicing subcomplexes, which then form a supercomplex. The accumulation of Raa6 with splicing factors of the first reaction in the ΔtscA background can be explained by the annulled first splicing reaction.

The mutant Δraa6 is only deficient in the second reaction [143]. Thus, the interaction of Raa6 with Raa2 and Raa2-specific proteins seems to be an essential step during psaA-i2 splicing. In contrast, the formation of the supercomplex affects both splicing reactions, but seems to be non-essential for both excision processes, since splicing of the first intron is not defective in mutants with impaired psaA-I2 splicing and vice versa [114,126,142]. A supercomplex might function in coordinating both splicing reactions to ensure that both introns are excised efficiently, which should lead to faster accumulation of the mature mRNA. In addition, this interaction could be important for recruiting the soluble subcomplex I to the thylakoid membrane. It is postulated that the mature psaA mRNA is directly translated at these membranes (see next section; [127]). Recent investigations revealed that formation of supercomplexes also occurs during splicing of nuclear mRNA introns [159]. These supraspliceosomes contain up to four active spliceosomes in a spatial arrangement and are connected by the pre-mRNA [160].

Overall, TAP results of the above study provide evidence that the splicing machinery not only requires RNP formation, but also depends on dynamic interactions [142]. Such a mechanism of dynamic, RNP-dependent association of catalytic structures is also used during splicing of spliceosomal introns. snRNAs are bound by protein factors and form snRNPs, which then associate and dissociate with the pre-mRNA to bring the catalytic RNA elements into close proximity [29]. These similarities suggest that the nuclear spliceosome and especially the splicing apparatus of *trans*-spliced group II introns have coevolved related mechanisms for the excision of introns.

## 14. Splicing of psaA-i2 Is Mediated by a Membrane-Associated Complex

Splicing factor Raa1 as well as splicing factors Raa2 and Raa7 are presumed to be attached to thylakoid membranes [111,127] (Figure 4). TAP–MS experiments with Raa2 and Raa7 support these observations, since purification of subcomplex II required addition of the detergent dodecyl-β-D-maltoside. Precisely how these proteins are attached to the thylakoid membrane is still unknown. For Raa2, it was shown that low concentrations of ammonium sulfate were sufficient to release it from membranes [127]. In contrast, the same concentration of ammonium sulfate was insufficient to purify subcomplex II via TAP. These results indicate that subcomplex II might be anchored in the thylakoid membrane by a transmembrane subunit, although in silico tools failed to predict transmembrane domains in subcomplex II proteins.

The membrane association of the psaA mRNA, encoding a subunit of photosystem I, indicates its translation at the thylakoid membrane [127]. Thus, splicing of psaA would simultaneously recruit the mature mRNA to the location of translation, avoiding transport of the mRNA or protein within the chloroplast. In fact, about 50% of the bacterial-like 70S ribosomes in *C. reinhardtii* plastids were shown to be membrane-bound [161,162]. In addition, several RNA-binding proteins with functions in translation are associated with low-density membranes [122,163,164]. For example, this was shown for Tab2, which is required for translation of psaB mRNA [165]. Even nonmembrane proteins, such as the RubisCO subunit RbcL, and the elongation factors EF-Tu, are translated at membrane-bound ribosomes [166,167].

A putative connection between the psaA splicing reaction and translation is also supported by TAP–MS analyses described previously [143]. Plastid ribosomal proteins S16, S17, and S20 were copurified with Raa2 and Raa7. All of these ribosomal proteins have orthologs in bacteria and were shown to be part of the small ribosomal subunit in chloroplasts of *C. reinhardtii* [168].

## 15. Putative Splicing Factors Identified by Mass Spectrometry

Two helicases identified in TAP experiments with Raa6 and Raa2 (Cre01.g022350, Cre07.g349300) are further promising candidates for putative new splicing factors. Helicases play a major role as remodelers during splicing of nuclear mRNA introns, as was previously described for Brr2 [169,170]. Furthermore, helicases are able to unwind RNA structures and thus allow correct folding into a required structure [171,172]. Such processes might also be crucial for organellar group II intron splicing. Indeed, it was shown for land plants that helicases may participate in splicing of group II introns [47]. The helicase PMH2 functions in the splicing of several group II introns in mitochondria of *Arabidopsis*, but seems only to play an enhancing, and not an essential, role [125]. Thus, it would be interesting to know whether both helicases identified above have similar functions during psaA intron splicing.

Furthermore, a factor related to the U3 small nucleolar RNA-associated protein 21 (Utp21) was identified with Raa2 and Raa6 as bait (Cre10.g466250). Utp21 is part of the SSU (small subunit) processome, which is required for biogenesis of the cytosolic 18S rRNA [173]. Only the C-terminus of Cre10.g466250 shows homologies to the human Utp21. BLAST analysis showed that additional homologs of Utp21 are not encoded in the nuclear genome of *C. reinhardtii*. Probably, Cre10.g466250 functions dually in the processome and psaA splicing reaction.

Remarkably, a splicing factor with homologies to maturases has not yet been identified in *C. reinhardtii*, and both psaA introns lack an intron-encoded ORF. The proteins discussed above that were identified using the TAP approach show no homologies to maturases. This raises the question of whether *C. reinhardtii* has completely lost its maturases. From an evolutionary point of view, this fact would be surprising, since a maturase homolog, Prp8, exists even in the nuclear spliceosome [60] (Figure 3). Organellar maturases in land plants are general splicing factors targeting many different group II introns [28]. However, the only known general factor of the psaA splicing machinery, Raa1, lacks any homologies to maturases; however, it should be noted that a further class B mutant defective in both splicing reactions does exist, namely L118B [141]. The locus affected is still unknown and may harbor a maturase gene.

## 16. Conclusions

Characterization of an RNA *trans*-splicing protein complex in *C. reinhardtii* led to the identification of several putative chloroplast splicing factors. Further comparisons between organellar and spliceosomal splicing factors suggest several evolutionary routes leading to the conversion of a self-splicing ribozyme into a protein-dependent splicing apparatus. An important challenge for future studies will be identifying further splicing factors and analyzing their specific contributions to the splicing process, such as RNA folding, RNA stabilizing, or remodeling of interactions. Their discoveries will provide further evidence for homologies between group II and spliceosomal introns. Several projects are underway to generate huge mutant libraries from *C. reinhardtii*, to collect more defects in photosynthesis or other essential developmental processes [174,175,176,177]. A genome-wide, indexed library of mapped insertion mutants from *C. reinhardtii* [177] has already been used to identify some, but by far not the majority, of the subunits of the chloroplast RNP supercomplex [143]. It can be envisaged that functional analysis of mutant strains will complete our view of a chloroplast spliceosome and contribute to our understanding of the relationship between organellar and nuclear spliceosomes.

## Figures and Tables

**Figure 1 cells-10-00290-f001:**
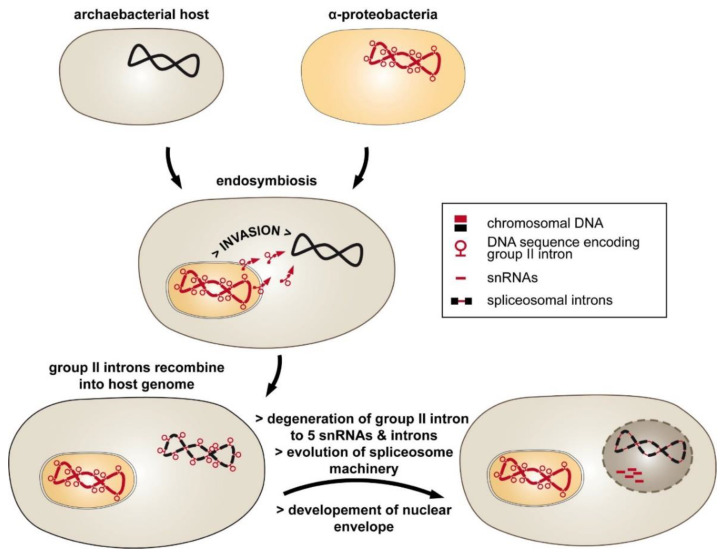
The introns-late theory explains the evolution of spliceosomal introns. The endosymbiosis of a group II intron-rich α-proteobacteria into the archaeal intronless host was followed by the invasion of the host genome by mobile group II introns. The resulting discontinuous genes provoked a strong selective pressure toward evolving intron removal. This included the degradation of group II intron RNA into mRNA introns and small nucleolar RNAs (snRNAs). Separation of the inefficient splicing reaction from translation was achieved by developing a nuclear envelope.

**Figure 2 cells-10-00290-f002:**
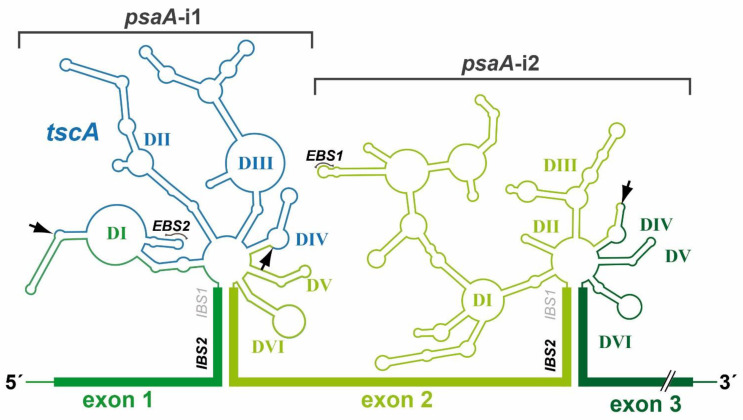
Secondary structure of psaA-i1 and psaA-i2 intron RNAs from *C. reinhardtii*. The three exons of the psaA gene and tscA locus of *C. reinhardtii* are transcribed independently. Two group II introns are formed at the exon boundaries of psaA by RNA base pairing. The mature psaA mRNA is generated by two *trans*-splicing reactions. Both group II introns have a characteristic group II intron structure, comprising six helical domains (DI-DVI), which surround a central core. psaA-i1 is a tripartite intron, whereas psaA-i2 is a dipartite intron. Fragmentation sites are indicated by arrows. Abbreviations: EBS/IBS, exon-/intron-binding sites.

**Figure 3 cells-10-00290-f003:**
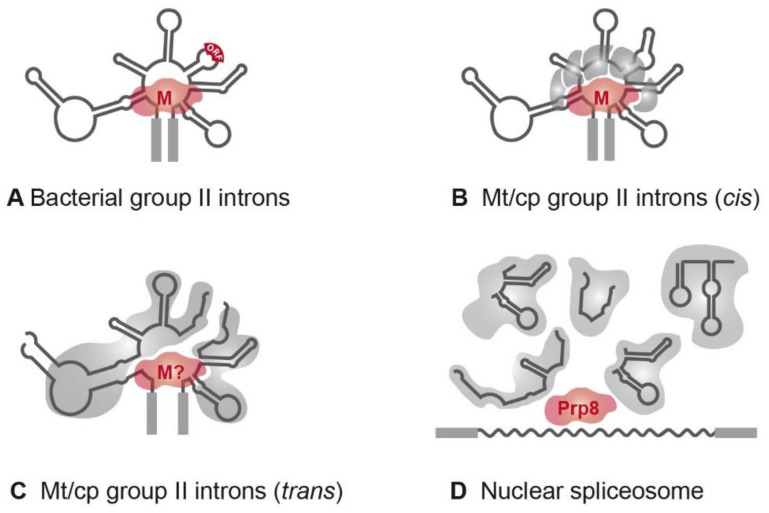
RNPs of group II and spliceosomal introns. (**A**) The majority of bacterial group II introns form an RNP with the intron-encoded maturase during the splicing reactions [14]. (**B**) In organelles, group II introns are degenerated and RNPs comprise at least five splicing factors [20]. The maturase is either encoded by an organellar group II intron (matR, matK) sequence or is nucleus-encoded (nMat1–4; [28]). (**C**) Fragmented group II introns of *C. reinhardtii* depend on complex RNPs comprising up to ten splicing factors and the precursor RNAs. A maturase homolog has not been identified yet. (**D**) The five *trans*-acting snRNAs of the nuclear spliceosome probably originated from fragmentation of group II intron sequences. snRNAs associate with a large number of protein factors to form a complex with snRNPs and function in splicing of nuclear mRNA introns (reviewed in [4]). Homologies to group II intron maturases were shown for the splicing factor Prp8 [60]. Abbreviations: M, maturase; Mt, mitochondrial; cp, chloroplast.

**Figure 4 cells-10-00290-f004:**
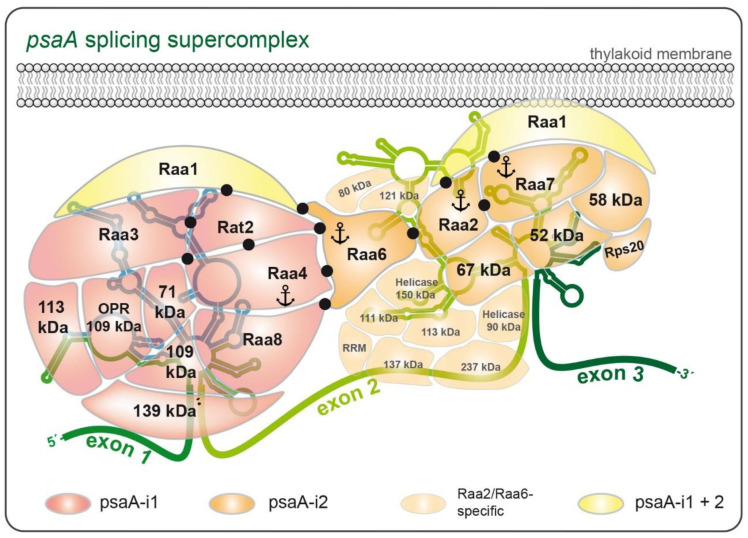
Model of subcomplex I and II involved in psaA *trans*-splicing. Comparative TAP analyses revealed that subcomplex I comprises at least ten core subunits. Five of these were previously characterized as splicing factors, namely Raa1, Raa3, Raa4, Raa8, and Rat2. Subcomplex I is an RNP and copurifies with psaA exon 1 precursor and tscA RNA [123]. Subcomplex II participates in the second psaA splicing reaction and comprises at least seven subunits [143]. These were identified by TAP technology, using Raa2 and Raa7 as baits. Copurification of psaA exon 2 and exon 3 precursors indicate that the membrane-associated (thylakoid membrane at the top) subcomplex II is able to form an RNP. Bait proteins used in this study for TAP are denoted with an anchor. Proteins involved in both psaA splicing reactions are highlighted in yellow, psaA-i1 splicing factors are colored in red, and psaA-i2 splicing factors are colored in orange. Uncharacterized proteins are denoted by their molecular mass in kDa. Subcomplex I: 71 kDa = Cre17.g724450; 109 kDa = Cre12.g533351; 109 kDa OPR = Cre01.g001501; 113 kDa = Cre08.g373878; 139 kDa = Cre11.g467652. Subcomplex II: 52 kDa = Cre03.g179000; 58 kDa = Cre17.g728850; 67 kDa = Cre02.g073200 (OPR, octatricopeptide repeat).

**Table 1 cells-10-00290-t001:** Functionally characterized *trans*-splicing factors from *C. reinhardtii*.

Abbreviation	Functional Domain	Size (kDa)	Involved in Intron *Trans*-Splicing	References
Raa1	OPR domains	210	psaA-i1 psaA-i2	[111]
Raa2	Pseudouridine synthase	45	psaA-i2	[127]
Raa3	No functional annotation	180	psaA-i1	[126]
Raa4	No functional annotation	116	psaA-i1	[142]
Raa6	No functional annotation	113	psaA-i2	[143]
Raa7	No functional annotation	130	psaA-i2	[144]
Raa8	OPR domains	269	psaA-i1	[123]
Rat2	OPR domains	144	psaA-i1	[123,145]

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
