# Peer review of "The Chloroplast Trans-Splicing RNA–Protein Supercomplex from the Green Alga Chlamydomonas reinhardtii"

_cells, 2021, doi:10.3390/cells10020290_

Round 1

Reviewer 1 Report

The manuscript " The chloroplast trans-splicing RNA-protein supercomplex from the green alga Chlamydomonas reinhardtii" by U. Kück and O. Schmitt is an interesting and comprehensive review on trans-splicing, comparing nuclear spliceosomes and group II intron-mediated splicing mechanisms in organelles and their evolution. This culminates with a description of the composition of the splicing supercomplex of the psaA transcript in the gree algae Chlamydomonas reinhardtii. This review provides an up-to-date overview of the subject and should be of interest to scientists involved in the search of the splicing complexes in organelles of higher plants.  

The genes and transcripts should be in italics, as well as the latin names.

The english is very good and I only noted a few typos:

  • Line 29, "und" should be changed to "and"
  •  Line 361, "CSR2 (chloroplast RNA splicing 1)" should be changed to "CSR2 (chloroplast RNA splicing 2)"
  • Line 380, "a additional subgroup" should be "an additional subgroup"
  • Line 649, "dodecyl-β-D-maltosid" sould be changed to "dodecyl-β-D-maltoside"

Author Response

To the editor of “Cells”.

In the following, we have responded point by point to the reviewers’ comments. The responding text is given in bold face.

All figures are made by our self, thus no copy rights are related to a third party.

We hope that we have fulfilled all requirements for accepting our manuscript for publication.

We are looking forward to hearing from you,

Yous sincerely, Ulrich Kück.

Response to the reviewers’ comments:

Reviewer #1:

We appreciate the reviewer’s comment that we have provided an up-to-date overview, which is of interest of scientists in this scientific field.

All the typos mentioned by the reviewer were corrected.

The manuscript " The chloroplast trans-splicing RNA-protein supercomplex from the green alga Chlamydomonas reinhardtii" by U. Kück and O. Schmitt is an interesting and comprehensive review on trans-splicing, comparing nuclear spliceosomes and group II intron-mediated splicing mechanisms in organelles and their evolution. This culminates with a description of the composition of the splicing supercomplex of the psaA transcript in the gree algae Chlamydomonas reinhardtii. This review provides an up-to-date overview of the subject and should be of interest to scientists involved in the search of the splicing complexes in organelles of higher plants.  

The genes and transcripts should be in italics, as well as the latin names.

The english is very good and I only noted a few typos:

  • Line 29, "und" should be changed to "and"
  • Line 361, "CSR2 (chloroplast RNA splicing 1)" should be changed to "CSR2 (chloroplast RNA splicing 2)"
  • Line 380, "a additional subgroup" should be "an additional subgroup"
  • Line 649, "dodecyl-β-D-maltosid" sould be changed to "dodecyl-β-D-maltoside"

Reviewer 2 Report

This review by Kuck and Schmitt can be divided into two related pieces. The larger first half is a discussion of intron evolution and the evidence that bacterial group II self-splicing introns gave rise to spliceosomal introns in eukaryotes and the trans-splicing of separately transcribed RNA molecules in organelles. The smaller second half is a detailed summary of the components that comprise the machinery involved in the two trans-splicing events that produce psaA mRNA in the chloroplasts of Chlamydomonas. This review should be relevant to anyone interested in trans-splicing or introns in general.

This manuscript is generally very well-written but I do have a few comments and suggestions. The current title reflects only the material presented in the second half of the review. Because the article depends so heavily on comparisons to group II introns, it would be worth making sure that all readers remember what group II introns are by defining them early in the review and briefly explaining how they differ from the other types of introns mentioned on line 38. A little more ‘connective tissue’ would be helpful to relate the detailed description of the composition of the splicing complexes presented in the second half of the review to the discussion of intron evolution in the first. For example, the ideas in the paragraphs in lines 636 to 643 and the Conclusions section could be moved to the start of section 10 to emphasize why it is important to examine the proteins involved in splicing of those specific introns before getting into the detailed results. The ‘bulged nucleotide’ mentioned in line 112 is not explained until line 144. The sentence starting on line 528 is a little unclear. Does it mean that subcomplex IIA forms subcomplex IIB by associating with the RNA or are those two events separate?

The manuscript contains a few minor typos.

Line 29, “und” should be “and”.

Line 136, “The detection of intron is” needs either an “an” after the “of” or plural “introns”.

Lines 504 and 505, “kDa” should be “MDa”.

Author Response

To the editor of “Cells”.

In the following, we have responded point by point to the reviewers’ comments. The responding text is given in bold face.

All figures are made by our self, thus no copy rights are related to a third party.

We hope that we have fulfilled all requirements for accepting our manuscript for publication.

We are looking forward to hearing from you,

Yous sincerely, Ulrich Kück.

Response to the reviewers’ comments:

Reviewer #2:

This review by Kuck and Schmitt can be divided into two related pieces. The larger first half is a discussion of intron evolution and the evidence that bacterial group II self-splicing introns gave rise to spliceosomal introns in eukaryotes and the trans-splicing of separately transcribed RNA molecules in organelles. The smaller second half is a detailed summary of the components that comprise the machinery involved in the two trans-splicing events that produce psaA mRNA in the chloroplasts of Chlamydomonas. This review should be relevant to anyone interested in trans-splicing or introns in general.

This manuscript is generally very well-written but I do have a few comments and suggestions. The current title reflects only the material presented in the second half of the review. Because the article depends so heavily on comparisons to group II introns, it would be worth making sure that all readers remember what group II introns are by defining them early in the review and briefly explaining how they differ from the other types of introns mentioned on line 38.

We have considered this point and give a more detailed description of the different introns mentioned (line 54-59)

A little more ‘connective tissue’ would be helpful to relate the detailed description of the composition of the splicing complexes presented in the second half of the review to the discussion of intron evolution in the first. For example, the ideas in the paragraphs in lines 636 to 643 and the Conclusions section could be moved to the start of section 10 to emphasize why it is important to examine the proteins involved in splicing of those specific introns before getting into the detailed results.

This is a very helpful comment by the reviewer, which was considered by rewriting the beginning of paragraph 10 (line 439 - 445)

The ‘bulged nucleotide’ mentioned in line 112 is not explained until line 144.

We have deleted the sentence on line 130, because it is not necessary at this point.

The sentence starting on line 528 is a little unclear. Does it mean that subcomplex IIA forms subcomplex IIB by associating with the RNA or are those two events separate?

This part has been rewritten and should be clearer for the readers (Line 542-545). Indeed, subcomplex IIA forms subcomplex IIB by associating with the RNA.

The manuscript contains a few minor typos.

Line 29, “und” should be “and”.

Line 136, “The detection of intron is” needs either an “an” after the “of” or plural “introns”.

Lines 504 and 505, “kDa” should be “MDa”.

All the typos mentioned by the reviewer were corrected.